# Do positive and negative shocks of tourism development affect income inequality in a developed country?

**Ngoc Bui Hoang** [ORCID]*

Finance, Economic and Management Research Group, Ho Chi Minh City Open University, Ho Chi Minh City, Vietnam

* ngoc.bh@ou.edu.vn

## Abstract

Income inequality is an essential cause of violence, stagnant development, and political instability. This study will examine the positive and negative shocks in tourism development, and the distribution of the interaction between tourism development, economic growth, human capital, globalization, and income inequality will be discussed in Singapore, a developed and top-visited country. By adopting autoregressive distributed lag and non-linear autoregressive distributed lag approaches for panel data from 1978 to 2022, the results indicate an asymmetric cointegration among variables, and positive and negative changes in tourism development lead to decreased income inequality. More specifically, the asymmetric effect of tourism is found both in the short- and long-term, and positive shock has a greater impact than negative shock. At the same time, the findings also reveal that economic growth and globalization enhance, while human capital negatively affects income inequality in Singapore. These findings strengthen the belief of Singapore policy-makers and recommend several significant lessons for developing countries to promote tourism, sustainable development, and reduce income inequality.

**Data Availability Statement:** We confirm that all sources of data were presented in the paper. We do not own data, and we do not have any special access privileges that others would not have.

## 1. Introduction

Income inequality (IE) is a social problem, and it is considered an undesirable consequence of economic policies issued by the government over a long time. Unfortunately, it has increased at varied rates in almost all countries [1]. Income inequality develops depending on the government's perspective, level of awareness, and residents' opposition. The widening gap between the rich and poor might limit the opportunities of poorer people and, over time, might cause violence and social and political instability [2, 3]. Therefore, reducing income inequality and then wholly eliminating it is the desire of many governments and people worldwide. Thus, economic policies that help reduce income inequality are always sought and welcomed [4–6]. Tourist development (TO) is often expected to be an effective policy for alleviating poverty because it brings a multiplicity of benefits to a country, such as promoting infrastructure, attracting foreign direct investment, generating new jobs, and stimulating

**Funding:** The author(s) received no specific funding for this work.

**Competing interests:** The authors have declared that no competing interests exist.

cultural change, thereby it could help reduce income inequality [4, 5, 7, 8]. However, Qiao and Chen [9] and Carrascal Incera and Fernández [10] revealed that a significant part of the labor force in tourism is related to self-employment or family enterprises. Njoya and Seetaram [11] also noted that TO requires investment and management experience, and poor people do not have these inputs. In some cases, a significant portion of tourism revenue may leak out of the local economy through imports, foreign-owned businesses, or multinational hotel chains. This can limit the trickle-down effect of tourism revenue and exacerbate income inequality [4]. Notably, Kinyondo and Pelizzo [12] argued that tourism has long been seen as a source of social inequality in Kenya, while Khan, Bibi et al. [13] revealed that there is a relationship between TO, terrorism, and inequality in Pakistan. Therefore, a comprehensive understanding of the impact of tourism on IE and its consequences has profound implications for policy-makers in planning strategies that prioritize local participation, equitable distribution of tourism benefits, and protect the rights and livelihoods of the poor.

Unfortunately, the debate on whether tourism can help reduce inequality is still ongoing. Studies on this topic have different conclusions and are not definitive. For example, Uzar and Eyuboglu [14], Porto and Espinola [15] found that tourism development increases inequality, while Nguyen, Schinckus et al. [3], Fang, Gozgor et al. [6], Shahbaz, Solarin et al. [16] suggested that tourism can lead to fairer income distribution. Another study by Fang, Gozgor et al. [6] revealed that TO significantly negatively influences IE in developing countries and has an insignificant impact in developed nations. Consequently, some researchers are studying how tourism development affects income distribution by looking at different factors like income, tax, price, and risk. The goal of this exploration is to fully comprehend how tourism impacts economic and social aspects of a country. Some reasons are given to explain the impact of TO on IE, such as (*i*) tourism can reduce income inequality by creating jobs for the poorest workers. However, tourism development may also lead to higher income inequality by increasing demand for infrastructure, benefiting the wealthier population [6]; (*ii*) the relationship between tourism and income distribution depends on institutions. A better institutional environment and a suitable tax policy can boost tourism opportunities, and in turn, tourism activities can also contribute to creating a more favorable institutional environment [3].

Singapore presents itself as a captivating context for investigating this association. Analysis of the relationship between tourism development and income inequality is essential for Singapore for the following reasons. First, the tourism industry generated around 527,500 jobs and contributed SGD 52.5 billion to the economy in 2020 [17]. Singapore's tourism development is an enticing example for emerging countries, as it enhances per capita income. Singapore, currently ranked among the top 10 countries in the world with the highest income, boasts an impressive per capita income of $60,729 per year (at the fixed price of 2010). However, it is important to note that such prosperity does not necessarily address the prevailing income disparity between the affluent and the less fortunate within Singapore. Startling figures from a recent report on income inequality indicate that in 2010, the income gap between the higher and lower-income classes was $9,288. Shockingly, this gap has widened to $12,840 in 2020, highlighting a growing challenge of eradicating inequality for Singaporean administrators. Second, the significance of tourism section has long been identified, so the Singaporean government has consistently devoted resources to fostering the growth of this industry. At the same time, Singapore has a diverse and well-integrated culture. These characteristics explain Singapore's spectacular achievement in tourism development, nowadays, it is a top-visited market in the world. Therefore, an analysis of the impact of tourism development on income distribution in Singapore might provide a practical lesson for emerging countries in planning tourism development strategies. Due to the two above reasons, the study is conducted to answer the research questions below:

Does tourism development affect income distribution in Singapore?

Is it the symmetric or asymmetric impact? If the asymmetric influence is found, how do positive and negative changes in tourism development affect income inequality?

## 2. Literature review

### The interaction between economic growth and income inequality

A fundamental question in sustainable development policies is whether economic growth retards or accelerates income inequality. The pioneering study of Kuznets [18] predicted that there is an inverted U-shape relationship between economic growth and income inequality. Following this prediction, many scholars investigated whether, how, and to what extent economic growth could affect income distribution. However, the answer remains elusive. A study that found a negative relationship is the study of Braun, Parro and Valenzuela [19], who conducted research that evaluated the major prediction of their model in terms of the influence of income disparity on growth at various degrees of financial development. They discovered that greater income inequality is related to weaker economic development by utilizing pooled OLS, FEM, and REM approaches in 150 countries between 1978 and 2012. Furthermore, they discovered that as economies' financial development levels rise, such an influence becomes substantially less pronounced.

In another study, Royuela, Veneri and Ramos [20] examined the relationship between income inequality and growth in 15 OECD nations from 2003 to 2013. It integrates household survey data and macroeconomic statistics from 15 OECD nations to cover over 200 similar areas. Their econometric findings showed an overall negative relationship between disparities and economic development since the beginning of the economic crisis. The sort of urban structure influences this connection. Higher disparities appear to be more destructive to growth in regions typified by medium to large-sized cities, whereas small cities and rural areas appear to be less affected. Likewise, Breunig and Majeed [21] recently re-examined the relationship between inequality and economic development in 152 countries. The study employed GMM and discovered that inequality had a detrimental influence on growth from 1956 to 2011. They also discovered that when both poverty and inequality were taken into account, the negative impact of inequality on growth was concentrated in nations with high poverty rates. More precisely, they found that when poverty (daily income below US$ 1.9), the impact of inequality on economic growth is greater.

Some studies show a positive relationship between economic growth and income inequality. Scholl and Klasen [22] reviewed the inequality-growth link, focusing on the significance of transition (post-Soviet) nations. They discovered a positive relationship between inequality and growth in the total sample using FE, GMM, and IV estimation methodologies driven by transition nations. Brueckner and Lederman [23] researched the connection between inequality and GDP per capita growth. Using panel data from 1970 to 2010, the researchers discovered that increased income disparity boosted transitory growth in low-income countries while having the opposite impact in high-income nations. However, Benos and Karagiannis [24] investigated the link between top income inequality and growth in the United States due to physical and human capital buildup. They determined that increases in inequality did not affect growth by utilizing 2SLS and GMM on an annual panel of the United States-level data from 1929 to 2013.

Application to Singapore, a high-income and top-visited country, Agiomirgianakis, Serenis and Tsounis [25] noted that the effective timing of economic policies actions in Singapore's tourism industry is not consistent. Moreover, many previous studies [22, 26] confirmed that

an increase in income of different population strata is not uniform. The rich will get richer, while the poor tend to get poorer. Therefore, the hypothesis is postulated that:

*Hypothesis 1*: *Economic growth positively affects income distribution in Singapore.*

## The relationship between tourism development, globalization, human capital, and income inequality

Some studies showed that tourism contributes to increasing income inequality. In particular, Chi [27] indicated that the long-run link between tourist revenue and income inequality differs across industrialized and developing nations. In developing countries, income disparity worsens as tourist revenue grows, improves after the first turning point is achieved, and worsens again after the second turning point. Similarly, Oviedo-García, González-Rodríguez and Vega-Vázquez [28] revealed that tourist income has not relieved poverty and has obviously failed to lessen wealth disparity.

It is plausible that activities associated with tourism might have varying effects on economic disparity, to the point that it has been statistically demonstrated that domestic tourism can make income disparity worse, while foreign tourism can reverse the situation (Nguyen et al., 2021). In contrast to the common idea that tourism contributes to economic expansion and reduces poverty levels, Zhang's study suggests that more tourism leads to greater income disparity (Zhang, 2021), especially in developing countries (Alam & Paramati, 2016). However, Uzar & Eyuboglu (2019) and Li et al., (2016) assert that tourism helps alleviate economic inequality, and Turkey's tourism is substantiated to affect income distribution. In Raza and Shad's paper, with the proviso that the countries make the expansion of their tourism industries a priority, they will be able to reduce the economic inequality that exists in the region (Raza & Shah, 2017). Thus, it is necessary to identify whether Singaporean tourism has a symmetric or asymmetric influence on its income inequality. Based on the above studies, the hypothesis is given as:

*Hypothesis 2*: *Tourism development reduces income inequality in Singapore*

One of the reported advantages of globalization is that it encourages the development of new employment, which, in turn, leads to an improvement in the standard of living of people all over the world and a reduction in poverty [29]. While Ghosh and Mitra [30] indicate that the distribution of income within highly developed countries is mostly unaffected by the growth of the tourism industry, [31] emphasize that more globalization causes a decline in the economic discrepancy that exists in that country. On the other hand, globalization magnifies the distribution of wealth by creating a skills gap in corporate operations, which in turn leads to a wider disparity in income [32]. By the same token, globalization has significantly widened the income gap between the rich and the poor in developing countries [6]. Therefore, globalization acts as a control variable in determining the relationship between tourism and income inequality.

According to social scientists, the term "human capital" refers to the attributes of a person that are deemed valuable to the production process. These attributes might include knowledge, skills, expertise, good health, education, and so on. The findings of Tsaurai [33] reverberate the findings of Becker [34], who noted that high levels of human capital development mean that individuals are talented, educated, and trained. They are also believed to be more productive, more marketable, and more likely to find a job that pays a higher salary. Therefore, investments in human capital development will allow developing nations to nurture a labor force of better quality and capable of maximizing earnings from tourism [30].

Some scholars confirmed the adverse effects of human capital and income inequality. Some examples include Ajide and Alimi [35], who found that the marginal effect of interactions between human capital measures and income inequality indicators is negative. Hu [2] pointed out the influence of income disparity on human capital inequality in 31 Chinese provinces, municipalities, and autonomous areas from 1996 to 2018. On the contrary, Chani, Janet al. [36] confirmed the presence of a long-run link and the causal relationship between human capital inequality and income inequality, and the Johanson cointegration and Granger causality tests are utilized. Suhendra, Istikomahet al. [37] used a panel data model with a fixed effect estimate for data from 34 Indonesia's provinces from 2013 to 2019 and found that human capital has a negative and significant effect on income inequality. According to the above researches and Table 1, the hypotheses are specified as:

*Hypothesis 3*: *Human capital contributes to reducing income inequality in Singapore*

*Hypothesis 4*: *There is a positive impact of globalization on income distribution*

Of course, none of the above studies and Table 1 could fully present the literature on the association between tourism development, economic growth, globalization, human capital, and income distribution. However, these studies also reveal that some natural issues have not been identified. One of the questions is whether the influence of TO on IE is symmetric or asymmetric. In Singapore, Khoi, Le and Ngoc [41] also indicated that the impact of TO on ecological footprint is asymmetric. Ozturk, Cetin and Demir [42] revealed that positive and negative income inequality shocks positively affect $CO_2$ emissions in Turkey. In addition, the asymmetric effect of other macroeconomic variables, such as financial development [26], foreign direct investment [48], energy consumption [49, 50], and exchange rate [51], could be found in previous studies. That means linear models may not be appropriate to explain the impacts of TO, and raise doubt about the effectiveness of suggested policies. From a methodological standpoint, an analysis including linear and non-linear frameworks gives a comprehensive understanding of the role and marginal effect of tourism shock on income inequality. Katrakilidis and Trachanas [43], and Ahmed, Zhang and Cary [44] pointed out that if the relationship between these variables is asymmetric, the policy implications based on the linear framework could be unreliable. Hence, this work aims to explore the role of tourism in reducing IE in Singapore by considering economic growth, globalization, and human capital in which tourism activities occur. The contribution of this study are summarized in detail:

i. Despite the abundance of previous studies on links between tourism and income inequality, the negative and positive changes in tourism development that might lead to effects of different magnitude or directions in income have been ignored. If the nexus between TO and IE is asymmetric, the policy implications based on symmetric methods can be unreliable. This study employs the linear ARDL and non-linear ARDL approaches to identify a symmetric or asymmetric effect of tourism on income inequality to provide a comprehensive understanding of tourism development in Singapore.

ii. Globalization and human capital are considered the keys to tourism development. However, the impact of two factors on income inequality is still fragmented and unclear in Singapore. In this study, globalization and human capital play as control variables in identifying the tourism-income inequality nexus.

iii. From 1978 to 2019, Singapore's economy suffered two financial crises (in 1997 and 2008), implying that the data might have some structural breaks. Therefore, an advanced unit root technique was also employed to probe structural breaks in the data, which is superior

**Table 1. A summary of available studies on the relationship between tourism development, economic growth, globalization, human capital, and income inequality.**

| No. | Authors | Duration | Countries/Areas | Econometric approach | Findings |
|---|---|---|---|---|---|
| **Tourism development–income inequality** | | | | | |
| 1 | Ofori, Dossou and Akadiri [38] | 1996–2020 | 48 African countries | GMM | + |
| 2 | Ghosh and Mitra [30] | 1995–2016 | 41 countries | FMOLS | Developing Countries:—Developed countries: + |
| 3 | Fang, Gozgoret al. [6] | 1995–2014 | 102 countries | Fixed-effects, FMOLS | Developing countries: - |
| 4 | Chi Chi [27] | 1995–2015 | 20 developed countries and 16 developing countries | FMOLS, DOLS | Developing countries: + |
| 5 | Nguyen, Schinckuset al. [3] | 2002–2014 | 97 countries | PCSE | - |
| 6 | Lv [39] | 1995–2012 | 113 countries | FMOLS | - |
| 7 | Shahbaz, Solarinet al. [16] | 1991–2017 | Malaysia | ARDL | - |
| 8 | Oviedo-García, González-Rodríguez and Vega-Vázquez [28] | 2000–2013 | Dominican Republic | ARDL | - |
| 9 | Porto and Espinola [15] | 2004–2015 | Argentina | FEM, REM | + |
| **Economic growth–income inequality** | | | | | |
| 1 | Breunig and Majeed [21] | 1956–2011 | 152 countries | GMM | - |
| 2 | Braun, Parro and Valenzuela [19] | 1978–2012 | 150 countries | Pooled OLS | - |
| 3 | Royuela, Veneri and Ramos [20] | 2003–2013 | 15 OECD countries | Pooled OLS REM | - |
| 4 | Scholl and Klasen [22] | 1961–2012 | 122 countries | FEM GMM | + |
| 5 | Brueckner and Lederman [23] | 1970–2010 | 144 countries, | 2SLS GMM | Low income countries: + High income countries: - |
| 6 | Benos and Karagiannis [24] | 1929–2013 | US state level data, | 2SLS GMM | No effect |
| **Human capital–income inequality** | | | | | |
| 1 | Ofori, Dossou and Akadiri [38] | 1996–2020 | 48 African countries | GMM | + |
| 2 | Ajide and Alimi [35] | 1980–2012 | 34 African countries | Poisson regression | + |
| 3 | Hu [2] | 1996–2018 | 31 provinces, China | Fixed effect, system GMM | + |
| 4 | Suhendra, Istikomahet al. [37] | 2013–2019 | 4 provinces Indonesia | FEM, REM | + |
| 5 | Lee and Lee [40] | 1980–2015 | East Asian | VECM | + |
| 6 | Chani, Janet al. [36] | 1973–2009 | Pakistan | VAR model, and Granger causality | + |

Notes: (+): positive impact; (-) negative impact.

to many other tests regarding size and accuracy in identifying breaks. Along with the results obtained from the non-linear ARDL approach, the findings of this study strengthen administrators' belief in promoting tourism and reducing income inequality in Singapore.

## 3. Methodology

Several factors affect the links between tourism—income inequality nexus, such as institutions [3], urbanization [45], human capital [46], and globalization [31]. Based on these studies, to identify the asymmetric effects of TO on income inequality in Singapore, our model is proposed in detail:

$$IE_t = \alpha_0 + \alpha_1.\ln TO_t + \alpha_2.\ln GDP_t + \alpha_3.HC_t + \alpha_4.Glob_t + e_t \tag{Eq1}$$

where, the $IE$ variable is income inequality (measured by Gini coefficient), and TO is measured by the number of international visitors. $GDP$, $HC$, and $Glob$ variables are economic growth, human capital, and globalization. Two variables ($TO$ and $GDP$) are transformed into natural logarithms to smooth data, while $IE$, $HC$, and $Glob$ are used as original data. Annual data is given by the Department of Statistics Singapore (for tourism, economic growth variable), Swiss Institute of Economics (globalization), and Federal Reserve Economic Data (human capital) from 1978 to 2022.

To examine the long-run association among the study variables, following the ARDL approach introduced by Pesaran and Shin [47], Eq 1 can be written as:

$$\Delta IE_t = \beta_0 + \beta_1.IE_{t-1} + \beta_2.\ln TO_{t-1} + \beta_3.\ln GDP_{t-1} + \beta_4.HC_{t-1} + \beta_5.Glob_{t-1} +$$
$$+ \sum_{j=1}^{p-1}\alpha_{1j}\Delta IE_{t-j} + \sum_{j=0}^{q}\alpha_{2j}\Delta\ln TO_{t-j} + \sum_{j=0}^{q}\alpha_{3j}\Delta\ln GDP_{t-j} + \sum_{j=0}^{q}\alpha_{4j}\Delta HC_{t-j} +$$
$$+ \sum_{j=0}^{q}\alpha_{5j}\Delta Glob_{t-j} + \varepsilon_t \tag{Eq2}$$

where, $\Delta$ represents the first difference, and $\varepsilon$ is the error term. $\beta_i$ ($i = 1,..,5$) are long-run coefficients, while $\alpha_i$ ($i = 1,..,5$) are short-run coefficients.

To probe asymmetric effects, the study follows the suggestion of Shin, Yu and Greenwood-Nimmo [48] to separate negative and positive changes in TO variable, as specific:

$$\ln TO_t = \ln TO_0 + \ln TO_t^+ + \ln TO_t^-$$

where, $\ln TO_0$ is the minimum number of international visitors, $\ln TO_t^+$ and $\ln TO_t^-$ are partial sum processes of the positive and negative changes in tourism, which are calculated as details:

$$\ln TO_t^+ = \sum_{j=1}^{t}\Delta\ln TO_j^+ = \sum_{j=1}^{t}\max(\Delta\ln TO_j, 0)$$

$$\ln TO_t^- = \sum_{j=1}^{t}\Delta\ln TO_j^- = \sum_{j=1}^{t}\min(\Delta\ln TO_j, 0)$$

Therefore, Eq 2 can be written as a non-linear ARDL(p,q) approach, as follows:

$$\Delta IE_t = \beta_0 + \beta_1.IE_{t-1} + \beta_2^+.\ln TO_{t-1} + \beta_2^-.\ln TO_{t-1} + \beta_3.\ln GDP_{t-1} + \beta_4.HC_{t-1} + \beta_5.Glob_{t-1} +$$
$$+ \sum_{j=1}^{p-1}\alpha_{1j}\Delta IE_{t-j} + \sum_{j=0}^{q}\alpha_{2j}^+\Delta\ln TO_{t-j} + \sum_{j=0}^{q}\alpha_{2j}^-\Delta\ln TO_{t-j} + \sum_{j=0}^{q}\alpha_{3j}\Delta\ln GDP_{t-j} +$$
$$+ \sum_{j=0}^{q}\alpha_{4j}\Delta HC_{t-j} + \sum_{j=0}^{q}\alpha_{5j}\Delta Glob_{t-j} + \varepsilon_t \tag{Eq3}$$

The null long-run asymmetric hypothesis is tested by $H_{0,LR}: -(\beta_2^+/\beta_1) = -(\beta_2^-/\beta_1)$ against the alternative hypothesis $H_{1,LR}: -(\beta_2^+/\beta_1) \neq -(\beta_2^-/\beta_1)$. Similarly, the null short-run asymmetric hypothesis is checked:

$$H_{0,SR}: \sum_{j=0}^{q} \alpha_{2j}^+ = \sum_{j=0}^{q} \alpha_{2j}^- \text{ against the alternative hypothesis } H_{1,SR}: \sum_{j=0}^{q} \alpha_{2j}^+ \neq \sum_{j=0}^{q} \alpha_{2j}^- \text{ The}$$

asymmetric cumulative dynamic multiplier effects of a unit change in tourism on income inequality are specified:

$$m_h^+ = \sum_{j=0}^{h} \frac{\partial IE_{t+j}}{\partial \ln TO_t^+}; m_h^- = \sum_{j=0}^{h} \frac{\partial IE_{t+j}}{\partial \ln TO_t^-} \text{ and } h = 0, 1, 2 \dots.$$

Note that as $h \to \infty$ then $m_h^+ \to \lambda_1^+, m_h^- \to \lambda_1^-$, where $\lambda_1^+$ and $\lambda_1^-$ are the asymmetric long-run coefficients calculated as $\lambda_1^+ = -(\beta_2^+/\beta_1)$ and $\lambda_1^- = -(\beta_2^-/\beta_1)$, respectively.

## 4. Empirical results

Based on the advantages of geographic location and many effective tourism development policies, Singapore is one of Asia's leading international passengers nowadays. In 2019, Singapore experienced a record number of visitors (attracting 19.1 million international visitors), following strong year-on-year growth in visitor numbers since 2015 [17]. Even so, Singapore has managed to market itself as a must-visit tourist destination in its own right, attracting repeat visitors by offering unique city experiences such as the Gardens by the Bay, and integrated resorts boasting casinos and amusement parks. The descriptive statistics of variables are shown in Table 2.

Table 2 shows that the highest value of income inequality is 39.30 percent, the human capital is 4.37 points, and the globalization index is 84.69 percent, but the lowest values in these three variables are 36.60, 1.65, and 62.78, respectively. The correlation matrix among variables is presented in Table 3. Accordingly, the correlation between *IE* and *HC*, *lnTO*, *lnGDP*, and *Glob* is 0.41, 0.44, 0,69, and 0,80, respectively, implying that there is a medium correlation between examined variables.

According to Pesaran and Shin [47], it is necessary to check the unit root before applying the ARDL approach because if any series is stationary at the second difference I(2), the estimated coefficients will not be valid. Moreover, another issue of time series data analysis is a structural break. To accomplish this, the augmented Dickey-Fuller [49], and Lee & Strazicich tests [50] were employed. Results in Table 4 show that all variables are stationarity at the first difference, and no variables are stationarity at the second difference, which implies the first condition to apply ARDL and non-linear ARDL approach is satisfied [51]. Furthermore, the Lee & Strazicich test provides break years. However, the results are inconsistent, and it could not help choosing a fit break year. Table 3 also indicates that the optimal lag length of *IE*

**Table 2. Descriptive statistic.**

| Variables | Obs | Mean | Std.Error | Minimum | Maximum |
|---|---|---|---|---|---|
| IE | 45 | 38.09 | 0.12 | 36.60 | 39.30 |
| lnTO | 45 | 15.61 | 0.13 | 11.39 | 16.76 |
| lnGDP | 45 | 10.38 | 0.08 | 9.38 | 11.12 |
| HC | 45 | 2.72 | 0.12 | 1.65 | 4.37 |
| Glob | 44 | 75.98 | 1.12 | 62.78 | 84.69 |

**Table 3. The correlation matrix.**

| Variables | IE | HC | lnTO | lnGDP | Glob |
|---|---|---|---|---|---|
| IE | 1 | | | | |
| HC | 0.410 | 1 | | | |
| lnTO | 0.441 | 0.269 | 1 | | |
| lnGDP | 0.695 | 0.931 | 0.431 | 1 | |
| Glob | 0.804 | 0.849 | 0.502 | 0.965 | 1 |

Unit root tests

variable is three, while the optimal order of *lnGDP* and *HC* variables is two. Thus, the optimal ARDL model is ARDL(3,0,2,2,0).

## Cointegration test

The findings of the unit root test indicated that our data series is integrated. Thus, the linear and non-linear Bound test of cointegration was applied. The result of two tests in Table 5 reveals that the F-statistic value in Eq 3 (= 9.855) is greater than the value of upper critical Bound at the level of 1 percent significance (= 5.06). Likewise, the result of t-statistic test (= -6.73) confirms that the same conclusion is smaller than the critical value of t-statistic test (= -4.79). These outcomes in Table 5 reveal that the non-linear ARDL approach is best for our model, and applying the linear ARDL approach might provide biased results.

## The short-run and long-run symmetric and asymmetric impacts

The short-run and long-run linear ARDL findings are presented in Table 6, while the outcomes of non-linear ARDL are shown in Table 7. In the long term, the coefficient of *lnTO*

**Table 4. Results of the unit root test and structural breaks test.**

| Variables | ADF test | | Lee & Strazicich test | Break year | Optimal lag |
|---|---|---|---|---|---|
| | I(0) | I(1) | | | |
| IE | -2.481 | -7.493*** | -7.894*** | 1989 & 2004 | 3 |
| lnTO | -1.539 | -12.847*** | -5.302 | 1995 & 2003 | 0 |
| lnGDP | -2.265 | -6.745*** | -6.747*** | 1996 & 2008 | 2 |
| HC | -2.247 | -3.447* | -6.405*** | 1994 & 2007 | 2 |
| Glob | -0.765 | -5.842*** | -6.681** | 1999 & 2004 | 0 |

Note: The ADF test is chosen based on the Akaike Information Criterion with intercept and trend. ***, ** and * respectively denote significance levels of 1%; 5% and 10%.

**Table 5. Results of the cointegration test.**

| Result of the Bounds test | | Critical value of F-statistic | | | Critical value of t-statistic | |
|---|---|---|---|---|---|---|
| Test statistic | Value | Signif | I(0) | I(1) | I(0) | I(1) |
| F-statistic in Eq 2 | 3.45 | 10% | 2.45 | 3.52 | -2.57 | -3.86 |
| t-statistic in Eq 2 | -1.65 | 5% | 2.86 | 4.01 | -2.86 | 3.79 |
| F-statistic in Eq 3 | 9.855 | 2.5% | 3.40 | 4.36 | -3.12 | -4.34 |
| t-statistic in Eq 3 | -6.73 | 1% | 3.74 | 5.06 | -3.43 | -4.79 |

**Table 6. The short- and long-run symmetric impacts.**

| Dependent variable: ΔIE | | | |
|---|---|---|---|
| **Variables** | **Coefficient** | **Standard error** | **T-ratio [Prob]** |
| lnTO | 0.130 | 0.113 | 1.16 [0.109] |
| lnGDP | 0.535 | 0.225 | 2.37 [0.024] |
| HC | -0.309 | 0.158 | -1.94 [0.062] |
| Glob | -0.005 | 0.014 | -0.34 [0.737] |
| CointEq(-1) | -0.153 | 0.034 | -4.44 [0.000] |
| ΔIE(-1) | -0.053 | 0.134 | -0.39 [0.694] |
| ΔIE(-2) | 0.306 | 0.119 | 2.56 [0.015] |
| ΔlnTO | -0.187 | 0.023 | -8.14 [0.000] |
| ΔlnTO(-1) | -0.252 | 0.052 | -4.81 [0.000] |
| ΔHC | 0.202 | 0.402 | 0.50 [0.619] |
| ΔHC(-1) | -1.925 | 0.345 | -5.56 [0.000] |
| ΔGlob | 0.027 | 0.008 | 3.37 [0.002] |
| ΔGlob(-1) | 0.033 | 0.009 | 3.68 [0.001] |
| Intercept | -0.473 | 0.125 | -3.78 [0.000] |
| $R^2$ | 0.9507 | $R^2$-adj | 0.9368 |
| $\chi^2_{SC}$ | 0.844 [0.441] | $\chi^2_{FF}$ | 2.579 [0.120] |
| $\chi^2_{NORM}$ | 1.442 [0.486] | $\chi^2_{HET}$ | 0.732 [0.717] |
| CUSUM test | Stable | CUSUMSQ | Not stable |

obtained from the symmetric ARDL approach is insignificant, implying no relationship between tourism and income inequality in Singapore. However, the results obtained from the asymmetric ARDL approach indicate that the influence of tourism on income inequality is evident. More specifically, the estimated coefficients of positive changes ($lnTO^+$) and negative changes ($lnTO^-$) are -0.461 and 0.240, respectively. The long-run asymmetric Wald test in the lowest of Table 7 provides evidence to reject the null hypothesis ($W_{LR}$ = 23.29, *p-value* = 0.000).

Likewise, the estimated coefficient of $\Delta lnTO^+$ variable (= -0.316, p-value = 0.006) is significant, while $\Delta lnTO^-$ variable is not significant (= -0.083, p-value = 0.109), and the short-run asymmetric Wald test also rejects the null hypothesis ($W_{SR}$ = 6.21, *p-value* = 0.019). These findings confirm that the impact of tourism development on income inequality is the asymmetry in both the short- and long-run. More specifically, a 1 percent increase in tourism leads to a 0.461 percent decrease in income inequality. Similarly, a 1 percent decrease in tourism leads to a 0.240 percent decrease in income inequality [48]. Hence, the positive changes in tourism have a greater effect than negative changes. Fig 1 also confirms this conclusion.

Similarly, Table 7 indicated that the estimated coefficients of *lnGDP*, *HC*, and *Glob* variables are significant in the long term. Economic growth and globalization positively impact income inequality, while increasing human capital might decrease income inequality. In the short term, only *lnGDP* variable has a positive influence and significant (= 0.752, p-value = 0.012). The estimated coefficient of *CointEq(-1)* variable is -0.479 and significant, and all diagnostic tests in Table 7, such as serial correlation, functional form, normality, and heteroskedasticity, have been successfully satisfied. In addition, Figs 2 and 3 demonstrate that both the CUSUM and CUSUMSQ lines comfortably fall within the critical bounds at a significant level of 5 percent, implying that Eq 3 remains stable [52, 53]. When all conditions of the non-linear ARDL approach are met, this study ensures that the findings are reliable and relevant for further analysis and the development of policy recommendations.

**Table 7. The short- and long-run asymmetric impacts.**

| Dependent variable: ΔIE | | | |
|---|---|---|---|
| **Variables** | **Coefficient** | **Standard error** | **T-ratio [Prob]** |
| $lnTO^+$ | -0.461 | 0.183 | -2.51 [0.019] |
| $lnTO^-$ | 0.240 | 0.132 | 1.82 [0.074] |
| lnGDP | 1.757 | 0.397 | 4.42 [0.000] |
| HC | -0.670 | 0.135 | -4.95 [0.000] |
| Glob | 0.044 | 0.007 | 5.67 [0.000] |
| CointEq(-1) | -0.479 | 0.071 | -6.73 [0.000] |
| ΔIE(-1) | 0.157 | 0.111 | 1.42 [0.169] |
| ΔIE(-2) | 0.244 | 0.123 | 1.99 [0.057] |
| $ΔlnTO^+$ | -0.316 | 0.104 | -3.04 [0.006] |
| $ΔlnTO^+$(-1) | 0.394 | 0.192 | 2.05 [0.051] |
| $ΔlnTO^-$ | -0.083 | 0.050 | -1.66 [0.109] |
| $ΔlnTO^-$(-1) | -0.625 | 0.159 | -3.92 [0.001] |
| ΔlnGDP | 0.753 | 0.278 | 2.71 [0.012] |
| ΔlnGDP(-1) | -0.466 | 0.348 | -1.34 [0.192] |
| ΔHC | -0.189 | 0.453 | -0.42 [0.679] |
| ΔHC(-1) | -0.694 | 0.439 | -1.58 [0.126] |
| Intercept | -0.764 | 2.416 | -0.32 [0.754] |
| $R^2$ | 0.9753 | $R^2$-adj | 0.9596 |
| $\chi^2_{SC}$ | 3.640 [0.058] | $\chi^2_{FF}$ | 1.739 [0.199] |
| $\chi^2_{NORM}$ | 1.158 [0.560] | $\chi^2_{HET}$ | 0.454 [0.947] |
| $W_{LR}$ | 23.25 [0.000] | $W_{SR}$ | 6.21 [0.019] |

Note: The superscripts "+" and "-" denote positive and negative partial sums. $\chi^2_{SC}$, $\chi^2_{FF}$, $\chi^2_{NORM}$, $\chi^2_{HET}$ denote LM tests for serial correlation, functional form, normality, and heteroskedasticity. The value in brackets is the corresponding p-value, respectively.

## Results of the causality test

The findings from ARDL and NARDL approaches have demonstrated the short- and long-run impacts of tourism development, economic growth, globalization, and human development on income inequality in Singapore. However, information about the causal relationship between these variables is also essential in suggesting lessons and policies. So, the Toda and Yamamoto procedure is employed. According to Sankaran, Kumaret al. [54], Ha and Ngoc [55], the Toda-Yamamoto procedure yields robust estimates irrespective of the integration and cointegration properties of the variables. If the value of $\chi^2$ (Chi_square) in the Toda-Yamamoto test has a p-value less than 0.05, implying the null hypothesis is rejected. According to Table 8 and Fig 4, the study finds a uni-directional causality running from income inequality to tourism development. Similarly, there is a uni-directional causality running from income distribution to globalization, while a bi-directional causality between human capital and income inequality is also confirmed. Overall, these findings are in line with the outcome of regression analysis. Hence, it allows us to conclude that our findings are convincing to suggest policies.

## Discussion

The empirical results suggest that international visitors contribute to reducing income inequalities in Singapore. That means hypothesis 2 is accepted. This finding is in line with Njoya and

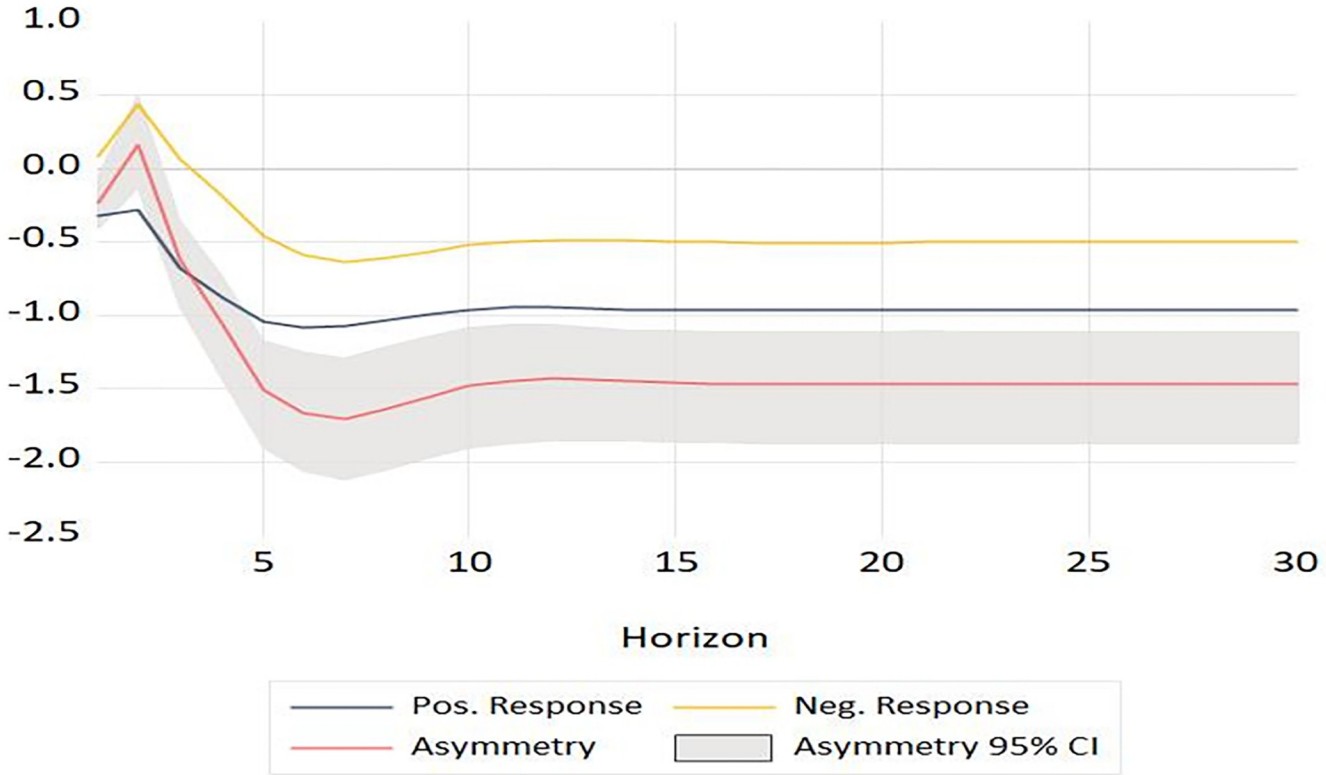

Fig 1. The shock evolution of tourism development on income inequality.

Seetaram [11] for Kenya, Li, Chenet al. [56] for China, and Uzar and Eyuboglu [14] for Turkey. However, it is inconsistent with Alam and Paramati [57]. The main contribution of this study shows that the influence of tourism on income inequality is asymmetric both in the short- and long-run. To explain for these findings, some reasons are given: (*i*) Singapore has an effective government and better institutions. It is easy to recognize that a dynamic government might create more opportunities for economic activities, such as tourism and financial services, while institutional security might attract more visitors [3]; (*ii*) Singapore has strong tax laws. Therefore, the tax evasion and tax avoidance acts are not fully valid in Singapore [58]; (*iii*) Singapore has a developed education system, giving individuals more jobs and a better income. However, a significant part of the labor force in the tourism industry in Singapore is involved in self-employment or family business, and tourism only benefits the owners. Hence, this might explain why the income inequality index of Singapore is not ranked the lowest in the world.

The positive coefficient of the *lnGDP* variable suggests that economic growth enhances income inequality in Singapore, implying that hypothesis 1 is also accepted. The finding is also similar to several studies, such as Scholl and Klasen [22], Benos and Karagiannis [24], and Elveren and Özgür [59]. In fact, Singapore has implemented several policies and initiatives to address income inequality and promote economic growth. For example, Singapore's government has reformed tax policies and applied progressive taxation. This tax system follows a progressive structure, with higher-income earners paying a higher percentage of taxes. Another

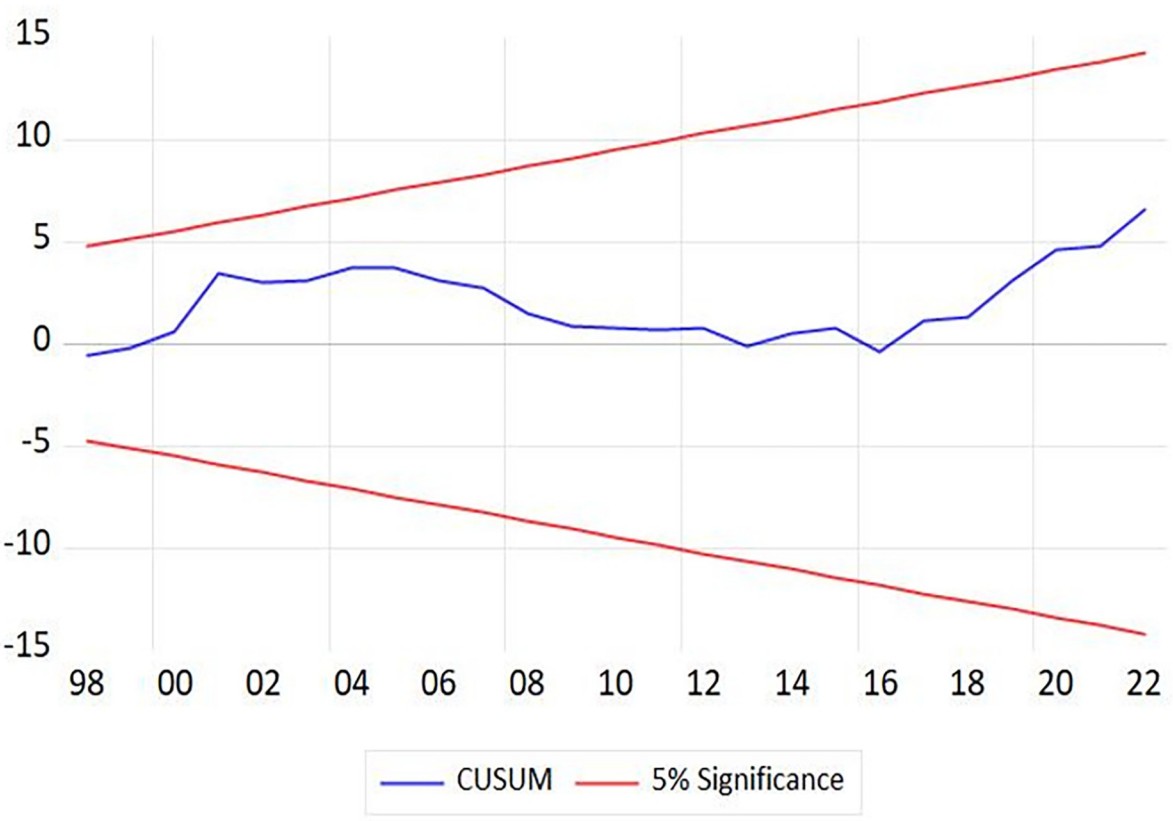

**Fig 2. CUSUM test.**

policy is the "SkillsFuture" program issued by the government, which provides training subsidies and support for individuals to upgrade their skills and stay relevant in the evolving job market. However, the increase in economic growth in the past few decades changed the income structure of individuals and classes. Despite the government's efforts, income inequality remains an ongoing challenge, and further measures may be needed to create a more equitable distribution of income and wealth in Singapore.

The finding also indicates that hypothesis 4 is accepted, and globalization positively drives income inequality in Singapore. This conclusion aligns with Haseeb, Suryantoet al. [60], Chishti, Ullahet al. [61]. Cristiano and İpek [62] noted that governance institution is a necessary condition to reduce the negative impact of globalization on inequality. In fact, Singapore's economy heavily depends on trade and foreign investment. So, Singapore's administrators have implemented policies and initiatives to create an attractive business environment, including favorable tax regimes, ease of doing business, and robust legal and regulatory frameworks to attract multinational corporations. Besides, Singapore's government has actively pursued free trade agreements with various countries and regions worldwide, emphasizing connectivity and innovation to enhance market access and reduce trade barriers, allowing Singaporean businesses to tap into global markets. However, globalization can widen income disparities within and between individuals [60] due to those with better access to education, technology, and capital are more likely to benefit from globalization, while those with limited access to such resources may be left behind.

Hypothesis 3 is also confirmed, and human development stimulates income equality in Singapore. This is not surprising since Singapore's human capital per person index has rapidly

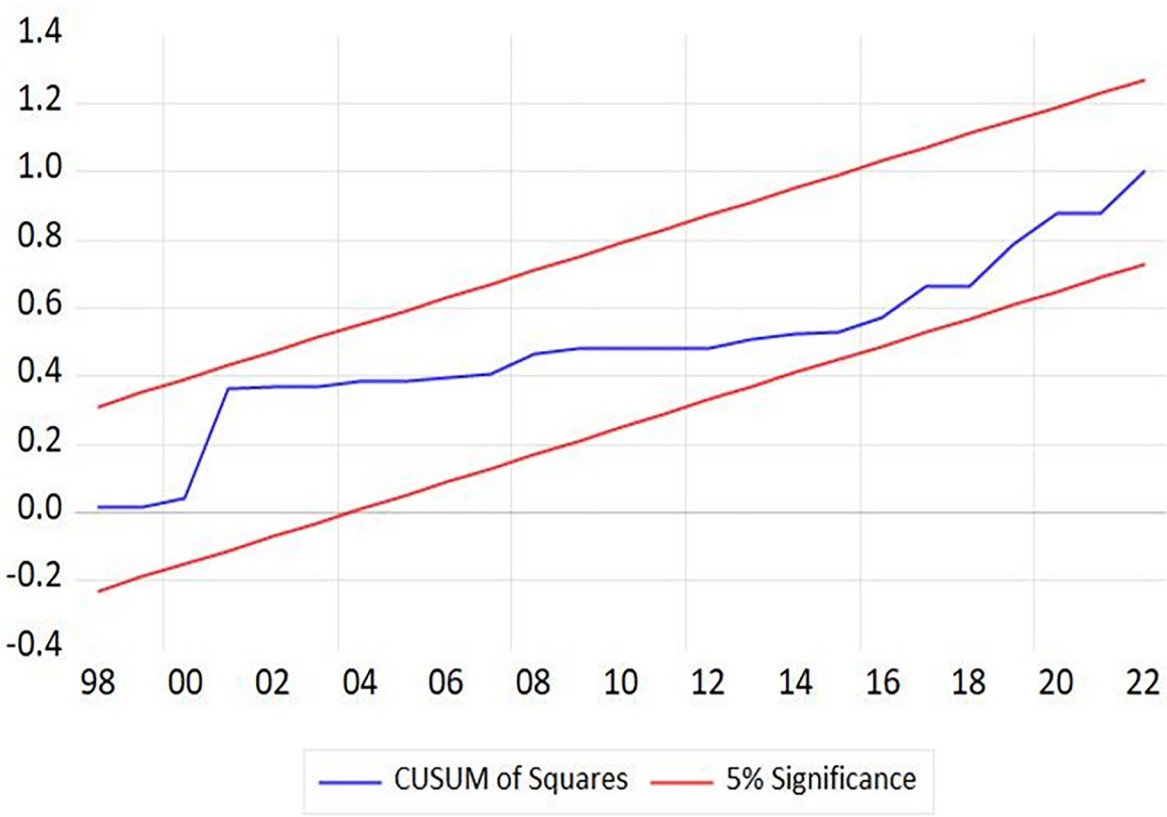

**Fig 3. CUSUMSQ test.**

increased along with GDP per capita. In the long term, Singapore has achieved high levels of income per capita, low unemployment rates, and a high standard of living. This country also has invested heavily in education, ensuring access to quality schooling for all citizens. Nowadays, Singapore has a well-developed public education system, which ensures holistic human development for all Singaporeans. Moreover, scholarships, bursaries, and financial aid programs are available to support students from lower-income backgrounds. This policy helps to reduce unequal access and improve income for individuals and workers.

## 5. Conclusion and policy implications

This study is conducted to probe the symmetric and asymmetric effects of tourism development on income inequality in Singapore, as well as economic growth, globalization, and

Table 8. Results of the Toda-Yamamoto test.

| Null hypothesis: No causality | Chi_square | p-value |
|---|---|---|
| lnTO does not Granger cause IE | 2.451 | 0.294 |
| IE does not Granger cause lnTO | 40.122 | 0.000 |
| lnGDP does not Granger cause IE | 1.522 | 0.467 |
| IE does not Granger cause lnGDP | 5.377 | 0.068 |
| Glob does not Granger cause IE | 5.766 | 0.056 |
| IE does not Granger cause Glob | 9.443 | 0.009 |
| HC does not Granger cause IE | 13.265 | 0.001 |
| IE does not Granger cause HC | 15.364 | 0.000 |

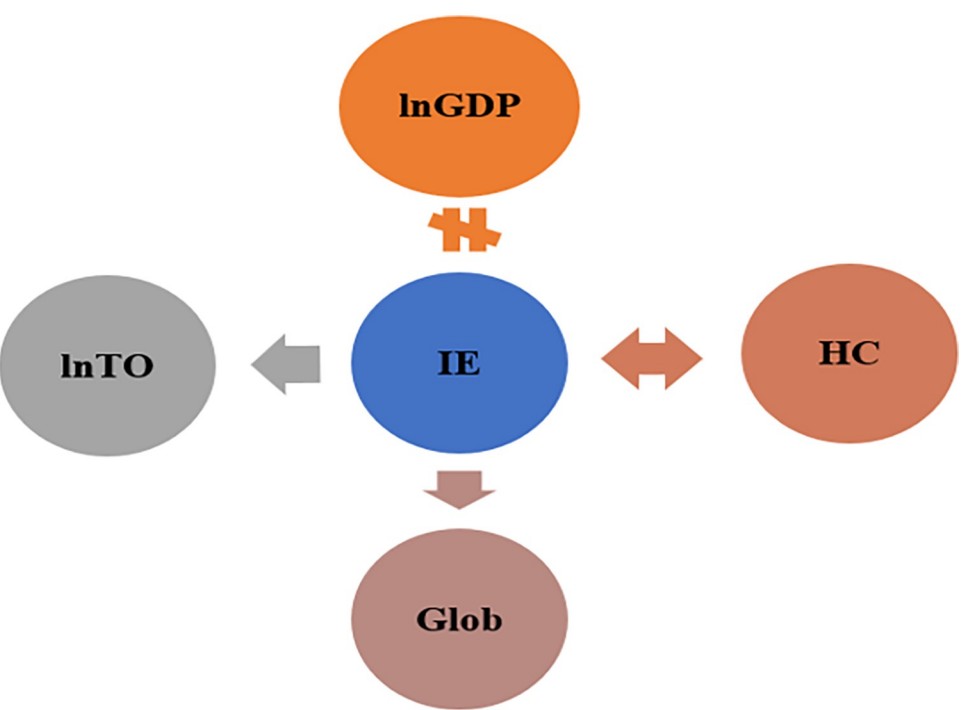

**Fig 4. The causality test results.** Note: →, ↔, and ≠ denotes a uni-directional, bi-directional, and no causality, respectively.

human development in the models. To achieve this goal, first, the study employed the unit root tests proposed by Dickey and Fuller [49], and Lee and Strazicich [50] with two structural breaks. Next, the Bound test technique introduced by Pesaran, Shin and Smith [51] was applied to assess the symmetric and asymmetric cointegration. Afterward, the symmetric ARDL method proposed by Pesaran and Shin [47], and the asymmetric ARDL approach given by Shin, Yu and Greenwood-Nimmo [48] were utilized to estimate the short- and long-run impact coefficients. Finally, the Wald test was adopted to conclude that the impact of tourism development on income inequality in Singapore is symmetry or asymmetry.

With a strict econometric strategy, our outcome provides some useful findings. *Firstly*, the study concludes that the impact of tourism on income inequality is asymmetric. More precisely, the impact of an increase in tourism development on income inequality has a larger effect than a decrease in tourism development. In addition, the causal test confirms a uni-directional causality from income inequality to tourism development. *Secondly*, the study also indicates that increasing economic growth and globalization might increase income inequality. *Thirdly*, improving human capital stimulates income distribution more equally in Singapore. Furthermore, there is a bi-directional causality running from human capital to income inequality.

## Policy implications

Based on the findings, the study delineates several significant policies that would considerably help Singapore's government reduce income inequality by developing the tourism industry, improving human capital, and maintaining sustainable development.

First and foremost, we suggest that policy-makers in Singapore should maintain appropriate policies to strengthen the tourism industry and encourage community-based tourism

initiatives. That means the government should empower local communities to participate in decision-making, provide training and capacity-building programs, and promote the development of small-scale enterprises and cooperative ventures. Besides, replacing outdated regulations and laws is necessary, especially in tax and transfer policies in tourism. Some policies, such as minimum wage regulations, collective bargaining rights, and workplace safety standards, also help reduce income inequality in Singapore. In addition, we suggest that Singapore's administrators focus on providing vocational training, promoting entrepreneurship, and fostering partnerships between tourism industry stakeholders and educational institutions. By equipping individuals with the necessary skills and knowledge, they can access higher-paying jobs within the tourism sector, reducing income disparities.

Finally, enhancing economic growth and expanding globalization must be accompanied by the improvement of human capital in order to mitigate the potential increase in income inequality. The government of Singapore should consider implementing social safety nets and welfare programs to reduce income inequality and provide support for vulnerable segments of society. These initiatives are essential to provide a safety net for individuals and families facing economic hardships, through targeted cash transfers, unemployment benefits, healthcare assistance, and other social services. By doing so, Singapore's government can ensure essential social protection for its citizens and mitigate the adverse effects of income inequality. These actions will lead to a more inclusive society and promote more equitable economic growth. The government can contribute to a fairer distribution of resources and opportunities by supporting those in need and reducing income disparities. It is important to note that the design and implementation of these programs should be tailored to the needs and context of Singapore, taking into account factors such as affordability, sustainability, and the overall impact on the economy.

Even though significant empirical evidence is acknowledged in this study, we agree that it still has some limits. In general, many macroeconomic variables can affect income distribution in Singapore. Thus, some related economic variables, such as foreign direct investment, institutional quality, and the development of the national education system, should be further considered. Furthermore, this research does not delve into the structure of variables (lead-lag variable) and heteroskedasticity impacts of tourism development through quantile analysis. Therefore, future studies should utilize novel econometric techniques such as the time-varying parameter vector autoregressive model and the quantile-on-quantile approach to provide a more comprehensive understanding of the role of tourism development, economic growth, and human capital in income distribution across various contexts. These recommendations are essential for supporting policy-makers in making informed decisions.

## Acknowledgments

We thank two anonymous referees and Editor-in-chief for their careful reading of our manuscript and their many insightful comments and suggestions that improved the quality of the original manuscript. Any remaining errors are our sole responsibility.

## Author Contributions

**Conceptualization:** Ngoc Bui Hoang.

**Formal analysis:** Ngoc Bui Hoang.

**Investigation:** Ngoc Bui Hoang.

**Methodology:** Ngoc Bui Hoang.

**Writing – original draft:** Ngoc Bui Hoang.

**Writing – review & editing:** Ngoc Bui Hoang.

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
