## [Decision Letter · Decision Letter 0]

8 Mar 2024

PONE-D-24-03256Do positive and negative shocks of tourism development affect income inequality in a developed country?PLOS ONE

Dear Dr. Bui Hoang,

Thank you for submitting your manuscript to PLOS ONE. After careful consideration, we feel that it has merit but does not fully meet PLOS ONE’s publication criteria as it currently stands. Therefore, we invite you to submit a revised version of the manuscript that addresses the points raised during the review process.

**ACADEMIC EDITOR: **

Dear Dr. Bui Hoang,

Thank you for your hard work on this manuscript. The reviewers found the technical aspects and analysis to be sound, which is great news. However, they've provided some important feedback to strengthen your paper. Here are the key points they'd like you to address in your revision:

Introduction:

Clearly lay out the specific research questions you are investigating based on the relationships you are exploring.Identify a gap in the existing literature that your study aims to fill, and explain how your work addresses that gap.Formulate hypotheses for each independent variable, grounded in theoretical literature.Shorten and tighten up the Introduction and Literature Review sections as suggested.

Methodology:

Justify why you analyzed data from the 1978-2019 period specifically.Include and describe a correlation matrix in this section or the Results.

Results & Discussion:

In the Discussion, compare and contrast your long-term findings in detail with the empirical results from other relevant prior studies.

Conclusion:

Provide a thorough discussion of the study's limitations.Offer recommendations for future research building on your work.

Additional References:

Consider incorporating the 3 additional references suggested by Reviewer 1.

Abstract:

Highlight the key findings more prominently in the Abstract section. ==============================

We look forward to receiving your revised manuscript.

Kind regards,

Abdullah Mohammad Ghazi Al khatib, Ph.D.

Academic Editor

PLOS ONE

Reviewers' comments:

Reviewer's Responses to Questions

**Comments to the Author**

1. Is the manuscript technically sound, and do the data support the conclusions?

Reviewer #1: Yes

Reviewer #2: Yes

2. Has the statistical analysis been performed appropriately and rigorously? 

Reviewer #1: Yes

Reviewer #2: Yes

3. Have the authors made all data underlying the findings in their manuscript fully available?

Reviewer #1: Yes

Reviewer #2: Yes

4. Is the manuscript presented in an intelligible fashion and written in standard English?

Reviewer #1: Yes

Reviewer #2: Yes

5. Review Comments to the Author

Reviewer #1: This study explores the asymmetric effect of tourism development on income inequality (measured by the Gini coefficient) in Singapore from 1978 to 2019. However, in order to improve the study, the following requests must be done:

1. In the introduction, research questions should be created by considering the relationships between

2. How the 1978-2019 period was determined should be explained in detail in methodology.

3. Hypotheses regarding each independent variable should be created based on the theoretical literature.

4. A literature gap should be created based on empirical literature and how the study fills this gap should be emphasized at the end of this section.

5. Correlation matrix should be created and described.

6. Each of the long-term findings should be compared separately with the empirical findings of other studies.

7) In the conclusion, the limitations of the study and future recommendations should be given in detail.

8) The addition of the following resources to the study is important for the improvement of the article:

a. https://doi.org/10.1007/s10668-021-01922-y

b. https://doi.org/10.1007/s11205-021-02641-7

c. https://d1wqtxts1xzle7.cloudfront.net/53788828/iib_dergisi-libre.pdf?1499430014=&response-content-disposition=inline%3B+filename%3DTHE_IMPACT_OF_ECONOMIC_GROWTH_AND_TRADE.pdf&Expires=1709721013&Signature=ChaGI8m6ftbIeaamRhnDkib~j995wF8slD9p~wK5PmKJdeIkwfRAbt2u0VQUuO1jYlG3yTiKzK9H5SOqLsVVcIADEpCan3xCyzO4M0imWMziM9gJtlOriAUU~AQICcVROv7VkunlNZ-3mH63dC6YgHl-EyOGAG3aOHdpxFahO6RQtCd87W4LKAOMgE0hOpqv0S7pK3WSd-wRK8x5hnb8~H4rKnWCLBXF63qnmXgxVnvueqrW4OOyKvMSw4gLpEd1LaHMRmK~1Xu6WFk1tyZpd36VrbDPNxhvor5IdtoPSSh17nWPQG~fX1JrW9-787dMS1JFgcXvNdKEBmTRtrZbMA__&Key-Pair-Id=APKAJLOHF5GGSLRBV4ZA

Reviewer #2: The article is well prepared ,but still some major corrections are required in manuscript. My corrections are given in attached file. The following point I need address here as well.

(1) abstract section need to highlight key findings of investigations.

(2) Introduction and review of literature needs to be shorten,

Thanks

6. PLOS authors have the option to publish the peer review history of their article (what does this mean?). If published, this will include your full peer review and any attached files.

Reviewer #1: No

Reviewer #2: No

---

## [Author Response · Author response to Decision Letter 0]

13 Mar 2024

Dear Editor-in-chief and Reviewers

The authors thank Prof. Abdullah Mohammad Ghazi Al khatib - the Academic Editor of Plos One journal, and the two esteemed reviewers for their valuable comments. We regret some weakness of the original manuscript. In this revision, The authors have tried to address every comment and suggestion. All modified parts are indicated in green in the new version of the manuscript. The authors sincerely hope that the revised manuscript meets the expectations of the Editors and Reviewers.

---

## [Decision Letter · Decision Letter 1]

20 Mar 2024

Do positive and negative shocks of tourism development affect income inequality in a developed country?

PONE-D-24-03256R1

Dear Dr. Bui Hoang,

We’re pleased to inform you that your manuscript has been judged scientifically suitable for publication and will be formally accepted for publication once it meets all outstanding technical requirements.

Kind regards,

Abdullah Mohammad Ghazi Al khatib, Ph.D.

Academic Editor

PLOS ONE

Additional Editor Comments (optional):

Reviewers' comments:

Reviewer's Responses to Questions

**Comments to the Author**

1. If the authors have adequately addressed your comments raised in a previous round of review and you feel that this manuscript is now acceptable for publication, you may indicate that here to bypass the “Comments to the Author” section, enter your conflict of interest statement in the “Confidential to Editor” section, and submit your "Accept" recommendation.

Reviewer #1: All comments have been addressed

Reviewer #2: All comments have been addressed

2. Is the manuscript technically sound, and do the data support the conclusions?

Reviewer #1: Yes

Reviewer #2: Yes

3. Has the statistical analysis been performed appropriately and rigorously? 

Reviewer #1: Yes

Reviewer #2: Yes

4. Have the authors made all data underlying the findings in their manuscript fully available?

Reviewer #1: Yes

Reviewer #2: Yes

5. Is the manuscript presented in an intelligible fashion and written in standard English?

Reviewer #1: Yes

Reviewer #2: Yes

6. Review Comments to the Author

Reviewer #1: Significant requests were sent to the authors in my previous review. In this context, the prepared answers are quite satisfactory. All the requests have been performed.

Thank you.

Reviewer #2: The authors has done all the correctios as per comments. So my decision is article is accepted in present form.

7. PLOS authors have the option to publish the peer review history of their article (what does this mean?). If published, this will include your full peer review and any attached files.

Reviewer #1: No

Reviewer #2: No

---

## [Editor Report · Acceptance letter]

2 Apr 2024

PONE-D-24-03256R1 

PLOS ONE

Dear Dr. Bui Hoang, 

I'm pleased to inform you that your manuscript has been deemed suitable for publication in PLOS ONE. Congratulations! Your manuscript is now being handed over to our production team.

Kind regards, 

on behalf of

Dr. Abdullah Mohammad Ghazi Al khatib 

Academic Editor

PLOS ONE